# Fabrication of Low-Cost Resistance Temperature Detectors and Micro-Heaters by Electrohydrodynamic Printing

**DOI:** 10.3390/mi13091419

**Published:** 2022-08-28

**Authors:** Salman Ahmad, Khalid Rahman, Taqi Ahmad Cheema, Muhammad Shakeel, Arshad Khan, Amine Bermak

**Affiliations:** 1Faculty of Mechanical Engineering, Ghulam Ishaq Khan Institute of Science and Technology, Swabi 23420, Pakistan; 2Mechanical Department, CECOS University, Peshawar 25120, Pakistan; 3Division of Information and Computing Technology, College of Science and Engineering, Hamad Bin Khalifa University, Doha 5825, Qatar

**Keywords:** electrohydrodynamic printing, inkjet, additive manufacturing, micro-fabrication, printed sensors, Resistance Temperature Detectors (RTDs), micro-heaters

## Abstract

EHD printing is an advanced deposition technology that is commonly utilized for the direct manufacture of electrical devices. In this study, meander-type resistive electrodes consisting of silver nanoparticles were printed directly on rigid glass and flexible polyethylene terephthalate (PET) substrates. High-resolution patterns of ≈50 µm linewidth were successfully printed on untreated surfaces utilizing a bigger nozzle of 100 µm inner diameter after improving the experimental settings. The manufactured electrodes were evaluated and used as Resistance Temperature Detectors (RTDs) and micro-heaters in a systematic manner. The temperature sensors performed well, with a Temperature Coefficient of Resistivity (TCRs) of 11.5 $\times 10^{-3}/{}^{\circ}C$ and 13.3 $\times 10^{-3}/{}^{\circ}C$, for glass and PET substrates, respectively, throughout a wide temperature range of 100 °C and 90 °C. Furthermore, the RTDs had a quick response and recovery time, as well as minimal hysteresis. The electrodes’ measured sensitivities as micro-heaters were 3.3 °C/V for glass and 6.8 °C/V for PET substrates, respectively. The RTDs were utilized for signal conditioning in a Wheatstone bridge circuit with a self-heating temperature of less than 1 °C as a practical demonstration. The micro-heaters have a lot of potential in the field of soft wearable electronics for biomedical applications, while the extremely sensitive RTDs have a lot of potential in industrial situations for temperature monitoring.

## 1. Introduction

Electronic applications have become more affordable, versatile, and integrable in a range of materials and wearables thanks to recent breakthroughs in printed electronics [1,2]. For instance, printed sensors can detect temperature [3,4,5], relative humidity [6,7,8], strain, and pressure [9,10,11]. Printed electronics have become more appealing for a wide range of applications due to their cost-effective mass fabrication capacity and comparatively low environmental implications compared to most additive manufacturing technologies. Printed electronics also have a number of advantages over traditional lithographic production methods, which include time-consuming and expensive vapor deposition stages [12]. Despite the fact that printing technologies enable low-cost direct deposition on the substrate in a single step without the use of masks, there are still a few challenges that must be addressed in order to realize high-performance sensor fabrication, such as the printing of high-resolution functional material patterns, pattern ink instability, and the mechanical strength of deposited patterns on various substrates.

To address these issues, electrohydrodynamic (EHD) printing technology is employed to pull the fluid from the printing nozzle using an increased electric field. EHD printing has gained popularity as a non-contact printing method capable of producing small patterns with greater nozzle diameters [13,14,15]. Electronic device fabrication is still one of the most common uses for EHD printing [16,17,18,19,20]. EHD printing produces high-resolution and homogenous patterns, resulting in improved device performance efficiency.

Temperature is a fundamental physical characteristic that varies spatially and temporally, and it continues to be one of the most recorded quantities, monitored by a variety of temperature sensors [21,22,23]. The most commonly reported temperature sensor is the Resistance Temperature Detector (RTD), which directly links temperature to resistance. It has the advantages of a quick response time, excellent precision, and ease of fabrication. The sensor interface’s design is primarily utilized to improve its sensitivity, linearity, and signal level [24]. The resistance change as a function of temperature in RTDs is essentially linear and can be approximated using Equation (1), below [25].
(1)$$R\left(t\right)=R_{o}\;\left\{\;1+\alpha \left(t-t_{o}\right)\right\}$$
where $\left(t-t_{o}\right)$ is the change in temperature, R(t) is the resistance at the final temperature, Ro is the initial resistance, and α is the material constant termed the Temperature Coefficient of Resistivity (TCR). The TCR value is a key criterion for evaluating RTD-based temperature sensors’ performance. Another important statistic to consider when evaluating the sensitivity of an RTD-based temperature sensor is the change in resistance to temperature change (ΔR/ΔT). In the recent literature, several materials such as silver, carbon, and metal oxides have been reported for RTD-based temperature sensing. For example, gold-based electrodes with a TCR of 0.0020/°C were printed using an aerosol jet and were primarily employed as heaters due to their efficient heating of 12 °C/mW on a polyamide substrate [26]. The gold electrodes were also used for temperature and humidity sensing, with temperatures and humidity sensitivities of 4.81 × 10^−3^/°C and 0.03 pF/percent RH, respectively [27]. A metal oxide thermistor with a wide temperature sensing range was also constructed and evaluated using an aerosol jet [28]. Among these materials, Silver is the most extensively used of these materials in RTD temperature sensing applications. For example, a temperature sensor based on silver was manufactured on Kapton substrates using inkjet printing and demonstrated resistance variations over a temperature range of 20 °C to 60 °C with a TCR of $2.23\times 10^{-3}/{}^{\circ}C$ [3]. Moreover, a high flexible silver inter-digital electrodes temperature sensor with a good TCR value of 3.75 $\times \;10^{-3}/{}^{\circ}C$ and a working temperature range of 28 °C to 50 °C was also reported, which was created by inkjet printing [29]. This sensor has a fast response and recovery time, as well as good bendability and durability. Researchers have also used inkjet printing to create silver temperature sensors on paper substrates with a TCR value of 0.0011/°C [30]. In another study, silver nanoparticle ink was inkjet printed on paper at temperatures ranging from 20 °C to 80 °C [31]. PEDOT:PSS and silver nanoparticles, both used as sensing materials, were printed on paper substrates and characterized at temperatures ranging from 30 °C to 45 °C [32]. The reported TCR values of silver and PEDOT:PSS were 9.389 × 10^−4^/°C and −0.0139/°C, respectively. Nanocomposites including carbon nanotubes mixed with polymers have also been employed as active materials for thermistor-type temperature sensors, in addition to pure materials [33,34,35]. A temperature sensor was printed using a hybrid reduced graphene oxide/silver nanoparticle ink [36]. Inkjet printing was used to create the meander-type structure, and the results were found to be superior to commercial sensors, with a sensitivity of 0.1086/°C over a temperature range of 30 °C to 100 °C. On a polyethylene terephthalate (PET) substrate created using an inkjet printing procedure, similar results were obtained with pure silver serpentine-structure electrodes [37]. To summarize, the research on printed RTDs reveals that these sensors have a limitation in that they either have a high TCR value or a large operating range. This issue must be addressed in order for printed RTDs to be widely adopted. Printing the electrodes with smaller line widths and spacing is one solution to this challenge. Because EHD printing provides a high resolution, it is a better option for fabricating these sensors with a high TCR value and a wide sensing range at a low cost.

Joules heating, which is the conversion of electrical energy to heat in terms of resistive losses, is used in the printed micro-heaters. $P=V^{2}/R$ is Joule’s law, where *P* is the power dissipation, *V* is the applied voltage, and *R* is the load resistance. Silver, like RTDs, is a common material in printed heaters [38,39]. Silver nanowires (AgNWs) were formed on PDMS film using vacuum filtration to create a micro-heater that reached a maximum temperature of 200 °C when 10 V was applied [40]. Another AgNWs-based micro-heater with a maximum temperature of 102 °C was created using an inkjet process on a hard glass substrate [41]. The micro-heater’s basic structure can be efficiently printed in high-resolution using EHD, considerably improving its heating performance [42].

EHD printing is employed to deposit meander-type resistive silver electrodes on rigid glass substrates and flexible PET films in this study. The printed electrodes are utilized for RTDs and micro-heater applications. The performance of RTDs on both substrates is satisfactory based on the sensitivity and TCR values for detectors. The detectors have a wide operating temperature range and a quick response and recovery time. The RTDs are used as a heating source and are integrated into the bridge circuit for signal conditioning. The study finds that fine printing of silver electrodes as temperature sensors on PET substrates enables impulsive temperature measurements, whereas glass substrates are preferable for stable and long-term readings. The reported printed silver electrodes, which were employed as micro-heaters, demonstrated remarkable sensitivity on both substrates.

## 2. Experimental Details

The resistive sensors were produced using the EHD printing approach described previously [13,43]. The schematic illustration of the EHD printing system is shown in Figure 1a. The ink flow arrangement, high voltage power supply, substrate base motion stages, and a vision camera are all part of the system. The ink flow is controlled by an air compressor with the help of a pressure regulator and a pressure switch. The voltage supply system is made up of a high voltage amplifier and a function generator. To print the predesign tool path, a moving stage accurately controls the substrate. The cone-jet transition of the printing ink is visualized using a high-resolution camera.

Photographs of the experimental EHD printing equipment are shown in Figure 1b. As an XY moving stage for substrate motion, two Linear stages were layered on top of each other. Thorlabs provided these linear stages (DDSM100/M), which were controlled using brushless servo controllers (KBD101) and KINESIS software. For the Z-axis mobility, a manual regulated slider (XR25/M series) was employed. For controlling the ink flow to a printing nozzle (inner diameter ≈ 100 µm), an air compressor was connected to a pressure regulator (SMC, IR1000-01, Tokyo, Japan) and a digital pressure switch (L&T, DPS-310RX, Mumbai, India). To power the EHD printing system, a high voltage amplifier (Matsusada, AMT-5B20-L, Shiga, Japan) was employed as a DC source. To enable drop-on-demand EHD printing, a function generator (Keysight, 33210A, Santa Rosa, CA, USA) was used to deliver pulse voltage to the system. The pulse waveform was monitored using an oscilloscope (Tektronix, TBS1052B-EDU, Beaverton, OR, USA). For the detailed viewing of the EHD cone-jet phenomenon, a monochromatic camera (DCC1545M USB 2.0 CMOS, thorlab, Ely, UK) was attached. Glass substrates (thickness ≈ 1 mm) and PET films (thickness ≈ 250 µm) were utilized for printing. The silver nanoparticle ink (Sigma Aldrich, 901083, Burlington, MA, USA) was put in an ultrasonic bath for about 15 min before printing. The printed sensors were thermally sintered at 150 °C for 30 min in a dry oven after printing, as advised by the manufacturer (Drawell, DHG-9030A, Chongqing, China). For electrical connections, conductive paste (EN-06B8) was used.

The substrate was cleaned with acetone to remove dust particles. The ink was placed in the bath sonicator before loading into a syringe barrel for printing. The substrate was fixed on XY linear stages and the required stand-off distance was adjusted by a manually operated regulated slider. The pressure regulator was used to supply the required printing pressure. The pulse voltages, frequency, and duty cycle were provided with the help of a function generator. After providing pressure and pulse voltages, wait for 30 s to stabilize the cone, and then operate the command as desired with a specific substrate speed. The camera was mounted to visualize the tip of the nozzle during the process. Printed patterns were sintered in a dry oven.

## 3. Results and Discussion

Figure 2 shows the working envelope of the silver nanoparticle ink, which defines the conditions under which the ink meniscus generates a stable cone-jet/drop, i.e., the combination of applied pressure and voltage at which the ink meniscus forms a stable cone-jet/drop. It is evident from Figure 2 that the lowest pressure and applied voltage for steady con-jet/drop formation are 3.85 kPa and 0.75 kV, respectively (left-bottom of the enclosed area). Additionally, the highest pressure and applied voltage are 20 kPa and 3.78 kV, respectively (right-top of the enclosed area). The pulse voltage and base voltage are represented by V_p_ and V_b_ in the drop-on-demand EHD printing as a pulse waveform description. Other EHD printing parameters, such as pressure, stand-off distance, frequency, duty cycle, and substrate speed, have an impact on printing quality and line width. Table 1 lists all of the optimized experimental parameters of the DOD EHD printing process in achieving fine deposition.

The optimized values reported in Table 1 are used to deposit the meander-type resistive electrodes shown in Figure 3. Figure 3a shows an image of fine-resolution printed electrodes (linewidth ≈ 50 µm) on untreated substrates (without the hydrophobic treatment of substrate and nozzle) with no discontinuities, employing a nozzle size of ≈100 µm (inner diameter). The smoothness and connectedness of the silver nanoparticle-based electrodes are confirmed by the Atomic Force Microscopy (AFM) and Scanning Electron Microscopy (SEM) pictures shown in Figure 3b,c. The width of the electrode pattern, 44.5 µm, is depicted in the SEM picture in Figure 3c. The purity of the silver nanoparticles following the printing and sintering process is further validated by the Energy Dispersive X-ray (EDX) spectroscopy of the printed electrodes, as shown in Figure 3d. The deposited films on substrates have an average peak value of around 79 nm, with a roughness of 18 nm, according to the AFM results. This lower roughness contributes to the enhancement in the performance of the sensor.

### 3.1. Resistance Temperature Detectors

Figure 4a,b displays a schematic illustration and photograph of a lab-built setup used to characterize the performance of printed RTDs. For temperature changes, a concentrated heat source with a hot air controller is positioned atop the RTD sample. For temperature monitoring, a K-type commercial thermocouple is inserted into the manufactured electrodes. This commercial thermocouple is interfaced with a computer via an open-source electronics platform (Arduino UNO) to record temperature measurements in order to evaluate performance. The printed electrodes are attached to a digital multimeter (GPS-8055C) for monitoring the corresponding resistance changes when the detector temperature varies. Temperature increments of 5 °C are used to calculate resistance values. Both the heat source and the K-type thermocouple are characterized at the same temperature. The resistances fluctuate linearly with temperature on both substrates but vary rapidly on PET films, as seen in Figure 4c. With a temperature shift of 100 °C, the resistance of a glass substrate changes from 334 Ω to 720 Ω (a difference of 386 Ω). Similarly, with a temperature increase of 90 °C, the resistance of PET films increases from 334 Ω to 734 Ω (a difference of 400 Ω). The interaction of nanoparticles is affected by temperature, which changes the resistance of printed electrodes. The setup was unable to record any more data on PET films above 110 °C because the electrodes degraded due to the extreme heat. The ratio of resistance changes with temperature variations is shown in Figure 4d. The printed RTDs’ sensitivities and TCRs are calculated from the plotted data and given in Table 2. As can be seen from these results, RTDs printed on PET films perform better in terms of heat sensing than those printed on glass substrates, but for stable and long-term readings, glass is preferred. This is because PET films are thinner and have lower heat losses than glass substrates [42]. The silver-based temperature sensors were compared to other silver-based inkjet printed temperature sensors, and the results are shown in Table 3. The reported sensor has better results compared to previous printed RTD sensors due to the high-resolution and uniform printing achieved by the EHD printing technique. By increasing and decreasing temperature to minimum and maximum limits, Figure 4e illustrates that RTDs printed on a glass substrate have smaller hysteresis errors for the printed sensor. A repeated heating–cooling cycle test also confirms the RTDs’ performance stability. Figure 4f shows that the RTDs’ response is consistent after repeated use (six times). This RTD response repeatability also indicates the reproducibility of the EHD printing process.

The printed RTDs are next evaluated for response and recovery time, which is the time it takes for the RTD to respond to a temperature increase generated by a heat source by reaching a higher steady-state temperature and then returning to base temperature when the heat source is removed. Figure 4g shows the RTDs’ response and recovery time (single cycle) on both the rigid glass substrate and PET film. Within 2 s, the steady-state temperature of the electrodes on both substrates was reached and detected, proving the versatility and speed of the sensor.

We also put the sensor to the test in terms of long-term durability. The resistance of the RTD sensor was measured with respect to room temperature (25 °C to 45 °C) and altered over the course of three months during fabrication and testing (April to June). Table 4 shows that the measurements are nearly the same, with less error, when compared to Figure 4c’s measured values. This confirms the reliability of the printed sensor.

### 3.2. Micro-Heaters

According to the Joule heating phenomenon, electrical resistance is directly responsible for heating in electronic devices. We used the manufactured silver electrodes as microheaters in another application. A DC voltage is applied to the printed electrodes via a variable power source, and their temperature is monitored using an infrared (IR) thermal imaging camera. Figure 5a shows the time-dependent temperature change of the sample heaters (sheet resistance ≈ 0.511 ${Ω}^{-1}$) under various applied voltages ranging from 2 to 15 V (after 15 s). For both the glass and PET substrates, the results reveal that the temperature changes linearly with voltage. The temperature of the heater on the glass substrate increased from 36 °C to 92 °C as a result of a total voltage shift of 15 V. The temperature of the heater on PET film varied from 36 °C to 104 °C by altering the voltage by 10 V. When the applied voltage exceeded 10 V, the silver electrodes on PET films were destroyed. The sensitivities of micro-heaters for glass and PET substrates were computed and determined to be 3.3 °C/V and 6.8 °C/V, respectively. Figure 5b illustrates the temperature variation on the glass substrate as a function of voltage after 45 s of application. With a total increase of 15 V and a sensitivity of 4.3 °C/V, the temperature ranges from 36 °C to 100.5 °C. This is the micro-heater’s maximum temperature, at which the silver line connection deteriorates.

The typical photograph and IR image (taken at 12 V) of the printed micro-heater on glass substrate are shown in Figure 5c,d, respectively. Figure 5e shows the temperature variation of the micro-heater on the glass substrate over time when different applied voltages of 6 V, 8 V, and 12 V were used. The response and recovery procedure is completed in less than 2 min, regardless of the input voltages, exhibiting the rapid response of the printed heater. The total power and power density necessary for the heaters to reach the target temperatures are shown in Figure 5f,g, respectively. The heaters are clearly operated at a low power density (1.83 W/cm^2^), which is mostly due to the high conductivity of the printed electrodes

### 3.3. Signal Conditioning

Because of their minimal resistance change due to self-heating, RTDs are commonly employed in bridge circuits. Figure 6 is a Wheatstone bridge circuit schematic including the printed RTD. The self-heating of the RTD occurs as a result of power dissipation by the RTD, resulting in inaccuracy in the readings. As a result, the dissipation constant value on RTD specification sheets is the power change of the RTD corresponding to a 1 °C temperature change. Using the following relationship, the self-heating temperature (ΔT) can be determined from the ratio of the power dissipation by an RTD (*P*) and the dissipation constant (*P_o_*) [45].
(2)ΔT=PPo

The dissipation constant for the printed RTD is computed to be 0.01235 W/°C using the data in Figure 5g for the glass substrate. According to the bridge circuit configuration shown in Figure 6, when a 5 V source voltage is applied and the resistance R_3_ > 450 Ω is maintained, the self-heating temperature is less than 1 °C. Table 5 shows all of the resistance requirements for maintaining a temperature change of 1 °C in the self-heating range of 5 V to 25 V. These resistance conditions are determined from Wheatstone bridge formulas and the power–temperature correlations of the printed RTDs. The results indicate that the printed RTDs have a lower self-heating temperature, which improves their sensing capability in circuits.

## 4. Conclusions

To summarize, EHD printing is used to make silver nanoparticle electrodes on rigid glass substrates and flexible PET films. Smaller pattern widths (minimum linewidth ≈ 50 µm) are successfully printed without special substrate treatment as compared to the printing nozzle diameter of 100 µm. The sensors’ performance efficiency improves as a result of the reduced printed width attained using EHD. As applications, the electrodes are utilized as RTDs and micro-heaters. The printed RTDs had high sensitivity and a wide temperature range of operation. The RTDs are effectively used in a Wheatstone bridge circuit for signal conditioning as a practical device demonstration, exhibiting great performance. Similarly, the produced micro-heaters had acceptable sensitivities on both substrates. These low-cost printed RTDs and micro-heaters, we feel, have a lot of potential in environmental monitoring and soft wearable electronics.

## Figures and Tables

**Figure 1 micromachines-13-01419-f001:**
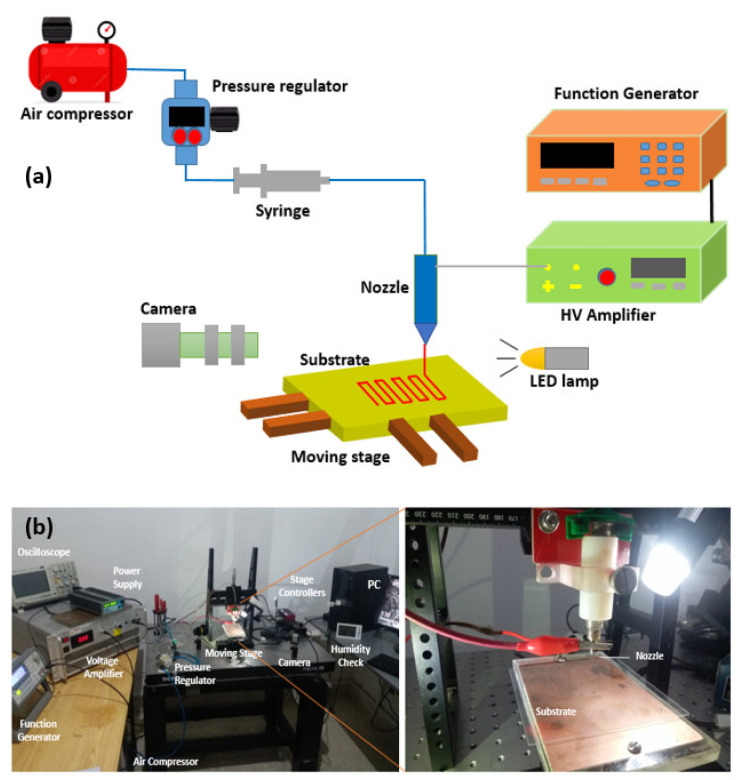
Electrohydrodynamic (EHD) printing experimental setup. (**a**) Schematic illustration. (**b**) Photograph.

**Figure 2 micromachines-13-01419-f002:**
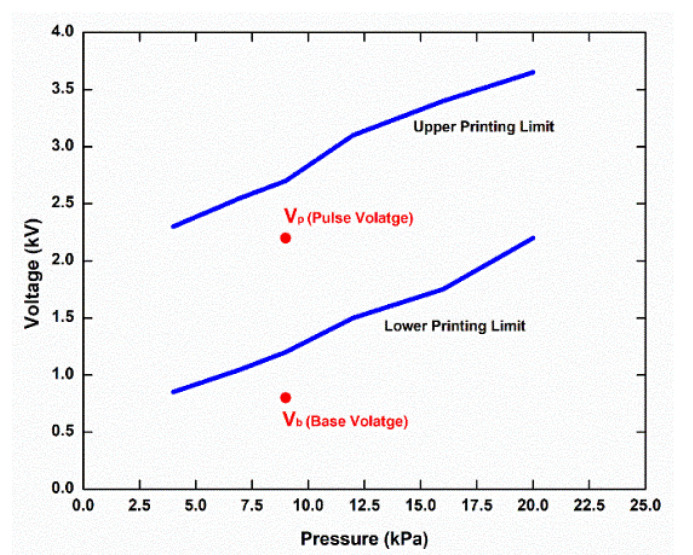
Silver nanoparticle ink operating envelope for steady cone drop/cone generation.

**Figure 3 micromachines-13-01419-f003:**
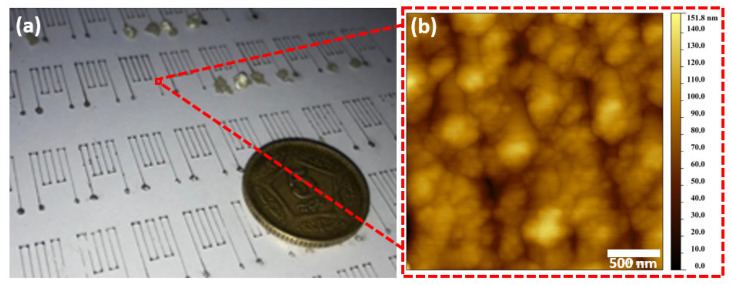
Morphological characterizations of the resistive silver electrodes on the glass substrate. (**a**) Photograph, (**b**) AFM micrograph, (**c**) SEM micrographs, and (**d**) EDX spectrum.

**Figure 4 micromachines-13-01419-f004:**
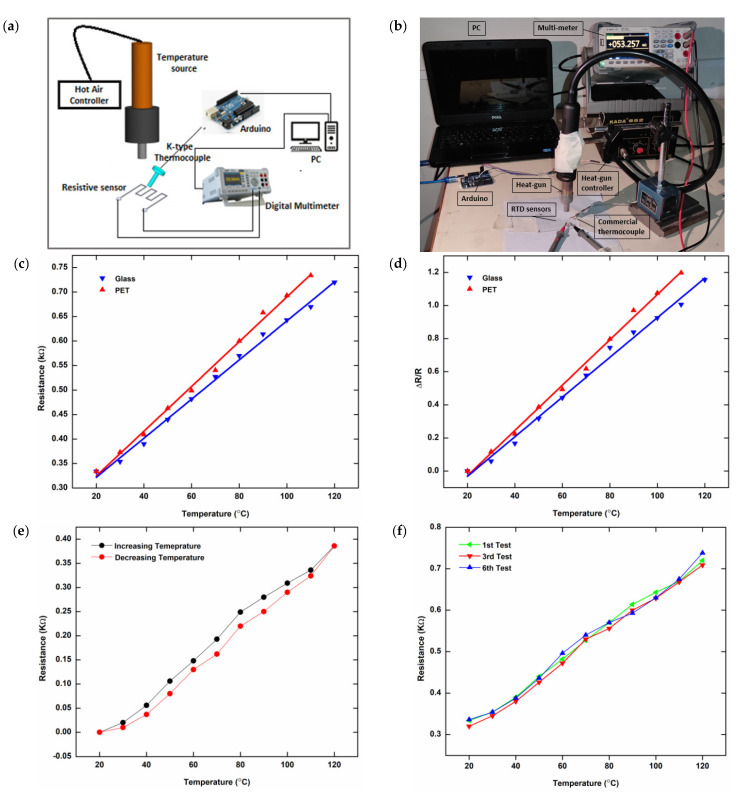
Characterization of printed RTDs. (**a**,**b**) Schematic illustration (**a**) and photograph (**b**) of the characterization setup for the temperature sensing; (**c**) resistance change with temperature var-iation; (**d**) normalized resistance change with temperature variation; (**e**) hysteresis graph of the sample (normalized data), printed on the glass substrate; (**f**) repeatability of the sample, printed on the glass substrate; and (**g**) response and recovery time, printed on both substrates.

**Figure 5 micromachines-13-01419-f005:**
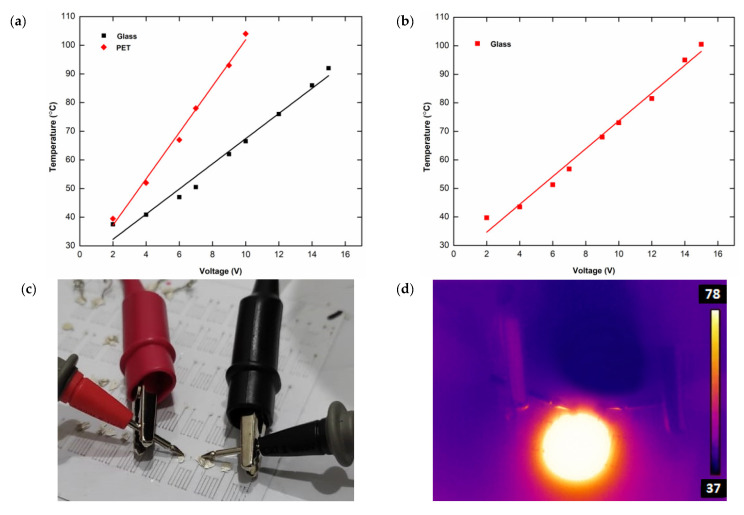
Characterization of printed microheaters. (**a**) Temperature–voltage relationship recorded after 15 s on both glass and PET substrates; (**b**) temperature–voltage relationship measured after 45 s on the glass substrate, printed micro-heater at 12 V, optical image (**c**) and IR image (**d**); (**e**) temperature variation with time at different applied voltages on the glass substrate; (**f**) temperature–power relationship; and (**g**) temperature–power density relationship on glass substrates.

**Figure 6 micromachines-13-01419-f006:**
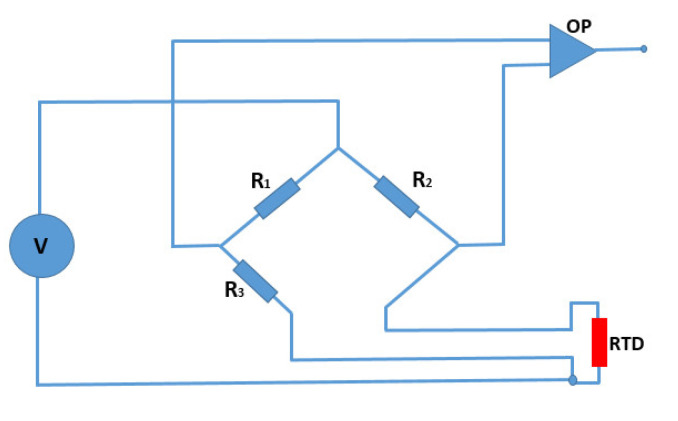
Wheatstone bridge circuit incorporating the fabricated RTD sensor.

**Table 1 micromachines-13-01419-t001:** Optimized experimental parameters for the EHD printing of silver nanoparticle ink.

Parameters	Values
Layers	2
Pulse Voltage	2.2 kV
Base Voltage	750 V
Pressure	9 kPa
Stand-off distance	1 mm
Speed	25 mm/sec
Frequency	700 Hz
Duty cycle	60 %

**Table 2 micromachines-13-01419-t002:** Performance characterization of the printed RTDs on rigid and flexible substrates.

Substrate	Glass	PET
Resistance change	386 Ω	400 Ω
Temperature change	100 °C (20–120) °C	90 °C (20–110) °C
TCR	11.5 × 10^−3^/°C	13.3 × 10^−3^/°C
Sensitivity	3.86 Ω/°C	4.44 Ω/°C

**Table 3 micromachines-13-01419-t003:** Comparison of silver-based RTDs inkjet printed on different substrates.

Material	Substrate	ΔT (°C)	TCR/Sensitivity
Ag [32]	Kapton	40	2.23 × 10^−3^/°C
Ag [33]	Kapton	22	3.75 × 10^−3^/°C
AgNPs [34]	Paper	40	1.1 × 10^−3^/°C
AgNPs [36]	Paper	21	9.389 × 10^−4^/°C
AgNPs/MoS_2_ [44]	Glass	100	2.46 × 10^−3^/°C
**AgNPs ^This work^**	**Glass**	**100**	**11.5 × 10^−3^/°C**
**AgNPs ^This work^**	**PET**	**90**	**13.3 × 10^−3^/°C**

**Table 4 micromachines-13-01419-t004:** Fabricated RTD values at different room temperatures, recorded over three months.

Weeks	Room Temperature (°C)	Avg. Resistance (Ω)	Measured Avg. Resistance [Figure 4c] (Ω)	Error %
1, 2	25	337	339	0.58
3, 4	30	348	354	1.69
5, 6	36	364	370	1.62
7, 8	40	387	390	0.7
9, 10	42	389	397	2.1
11, 12	45	407	420	3.09

**Table 5 micromachines-13-01419-t005:** Conditions for keeping the self-heating temperature change less than 1 °C in a Wheatstone bridge circuit.

Source Voltage (V)	Resistance Condition (kΩ)
5	R_3_ > 0.450
10	R_3_ > 1.3
15	R_3_ > 2.1
25	R_3_ > 3.7

## Data Availability

On behalf of all authors, the corresponding author declares that data supporting the findings of this study are available within the article.

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
