# Peer review of "Fabrication of Low-Cost Resistance Temperature Detectors and Micro-Heaters by Electrohydrodynamic Printing"

_micromachines, 2022, doi:10.3390/mi13091419_

Round 1

Reviewer 1 Report

Dear Authors, 

The manuscript is in good outstanding and addressed a fundamental issue on MEMS temperature detectors. The design and experimental implementation will make this manuscript one of the best of its kind. I have only one suggestion. Add details fabrication steps so the readers can follow and understand your procedure. 

Thank you. 

Author Response

The authors really appreciate valuable comments and review of this paper for improvement. Following your comments and suggestions, we have revised and improved the manuscript.

Reviewer 2 Report

The manuscript "Fabrication of Low-cost Resistance Temperature Detectors and Micro-heaters by Electrohydrodynamic Printing" by Ahmad et al., suggested a strategy for preparing Ag-based electrodes via electrohydrodynamic printing and their application to temperature detectors and micro-heaters. They found an optimized condition to print Ag nanoparticles with considering steady con drop/cone generation envelope, and demonstrated these printed Ag worked as active materials for temperature detecting. The subject is suitable for ‘Micromachines’, I recommend the major revision of this manuscript.

Question:

1.      There have been several research works to print Ag nanoparticles through electrohydrodynamic printing. What is originality of this work regarding EHD printing of Ag nanoparticles?

2.      The author decided an optimized stand-off distance of 1 mm. Is there systemic study of the printing behaviour depending on various stand-off distances?

3.      It would be better if the author reported electrical field parameter instead of applied voltage.

4.      Figure 3b AFM image: I think probe tip had some error. I believe the author can get more clear image of EHD printed Ag electrode.

5.      Is there any effect with humidity?

Author Response

(The authors gave the same response as above.)

Reviewer 3 Report

In this paper, EHD printing is used to make silver nanoparticle electrodes on rigid glass substrates and flexible PET films. Smaller pattern widths (minimum linewidth ≈ 50µm) are successfully printed without special substrate treatment as compared to the printing nozzle diameter of 100µm. The sensors' performance efficiency improves as a result of the reduced printed width attained using EHD. 

1. EHD technology is popular, and EHD pumps should be included in their background. some researchers use some EHD technology to realize the fluidic rolling robot using voltage-driven oscillating liquid and bidirectional electrohydrodynamic pump with high symmetrical performance and its application to a tube actuator. 

2. Figure 2 shows the lowest pressure and applied voltage for 157 steady con-jet/drop formations are 3.85 kPa and 0.75 kV, respectively. what is the meaning of Vb and Vp in the figure? 

3. The optimized values are reported in Table 1. I wonder how they can get these optimized values. the authors should provide some theory about the EHD printing technology if possible. 

4. Where did the author get the figure.3b, the authors should point to the position in the figure.3a where they capture the photo. And, the fonts in figure.3d are too small. 

5. Why is the resistance changing with the substrate in their devices? 

Author Response

(The authors gave the same response as above.)

Round 2

Reviewer 2 Report

Since the author improved the manuscript by reflecting reviewer's comment, I think this manuscript can be published in your journal. 

Author Response

Thank you for review process and highlighting the shortcomings in the manuscript for improvement. The reviewer’s recommendations are valuable and the authors appreciate their suggestions which helped in refinement of the submitted manuscript. 

The authors has read the manuscript for English editing.

Reviewer 3 Report

1. the authors should mark the Vp and Vb meanings in the figure.2 

2. Please consider the theory for your devices. I checked many EHD printing technologies, a small change will cause different results. The authors should consider the theory to confirm whether their results are convincing or not.  

3. Figure.3d is still blurred. 

Author Response

(The authors gave the same response as above.)

Round 3

Reviewer 3 Report

The authors can not provide some theoretical models.

I rest this in this time.